# Evaluation of the Effect of the Addition of Hydroxyapatite on Selected Mechanical and Tribological Properties of a Flow-Type Composite

**DOI:** 10.3390/ma15249016

**Published:** 2022-12-16

**Authors:** Zofia Kula, Leszek Klimek, Karolina Kopacz, Beata Śmielak

**Affiliations:** 1Department of Dental Technology, Medical University of Lodz, Pomorska Str. 251, 92-213 Lodz, Poland; 2Institute of Materials Science and Engineering, Lodz University of Technology, Stefanowskiego Str. 1/15, 90-924 Lodz, Poland; 3“Dynamo Lab” Academic Laboratory of Movement and Human Physical Performance, Medical University of Lodz, Pomorska Str. 251, 92-215 Lodz, Poland; 4Department of Prosthodontics, Medical University of Lodz, Pomorska Str. 251, 92-213 Lodz, Poland

**Keywords:** composite mechanical properties, composite tribological properties, dental composites, hydroxyapatite

## Abstract

(1) Background: The aim of the study was to determine the effect of modification with sintered hydroxyapatite (HAp) on selected mechanical and tribological properties of a flow-type composite. (2) Methods: Samples in the shapes of cuboidal beams (*n* = 120) and cylinders (*n* = 120) with the proper dimensions were prepared from a standard flow-type composite and others with the addition of 2% wt., 5% wt., and 8% wt. sintered hydroxyapatite. The bending strength, compression strength, diametral compression strength, impact resistance, hardness, and tribological properties were compared. (3) Results: In all cases, it was established that an increase in the amount of HAp caused a reduction in the bending, compression, and diametral compression strength. Increasing the amount of added HAp also reduced the impact strength, hardness, and wear resistance. However, the differences were statistically insignificant. (4) Conclusions: The addition of hydroxyapatite to a flow-type composite material worsened its mechanical and tribological properties; however, the obtained values were acceptable with 2% wt. and 5% wt. HAp.

## 1. Introduction

Dental composites are used to reconstruct a damaged tooth, thus improving its esthetics and restoring its chewing and speaking functions. These materials exhibit a similar structure to that of tooth tissue; they contain an organic substance (resin) and an inorganic substance (fillers) bound by a binding substance (vinyl silane) [1,2,3,4,5,6]. In addition to the resins (polymers, monomers), the organic phase also contains compounds regulating the polymerization process (initiators, inhibitors) in addition to substances conditioning the esthetic effects (dyes, UV absorbers, and others). The inorganic phase contains fillers based on silicon dioxide in the form of crystalline quartz and fillers made of glass, along with silicon-sodium, boron and lithium salts, and heavy metal oxides [1,6]. The fillers are added to strengthen the material and to reduce the shrinkability, radio-opacity, and thermal expansion coefficient. They are very hard and non-toxic. Fillers can be divided, depending on the particle size, into two basic groups: macrofillers and microfillers [3].

The advantages of composite materials include, *inter alia*, the possibility to choose an ideal filler color with a proper refractive index, good adhesion to glass, the possibility of fast tooth reconstruction and high crush resistance, possible cariostatic function, and contrast for X-ray pictures, which is especially important for the side lateral teeth [6,7]. However, such materials are subject to shrinkage during polymerization, leading to the formation of marginal gaps between the filling and the hard tissues of the tooth [5]. The shrinking mostly weakens the adhesive bonds, causing internal cracks in the material, increased porosity, and decreased mechanical strength [7]. Marginal gaps can cause microleaks, mostly a bacterial kind, which lead to discolorations on the filling’s circumference, secondary tooth decay, and most importantly, inflammation and necrosis of the pulp [5]. Temperature changes in the oral cavity result in the shrinkage and expansion of the filling materials, as well as the hard tooth tissue [7]. Ideally, the marginal thermal expansion coefficient of the materials and the tooth tissue should be similar.

Composite fillings must be, above all, biocompatible and durable, and thus, possess appropriate hardness, bending, and compression strength. In addition, they should not cause sensitivity of the teeth. So far, no ideal material for replacing hard tissue has been found. 

A good solution can be adding hydroxyapatite (HAp), an inorganic compound that is part of hard tooth tissue, bones, and pathologically calcified tissue, to the filler [6]. Chemically, it is a calcium orthophosphate, i.e., salts of tribasic orthophosphoric acid. The compound can be synthesized artificially. The chemical composition and the crystalline structure of synthetic hydroxyapatite are similar to those of natural hydroxyapatite. It is characterized by high biocompatibility and bioactivity. It does not cause inflammation or tissue irritation; it is also not toxic or carcinogenic [8,9,10,11]. It has been successfully applied in orthopedics and implantology, both in hard tissue regeneration and as an implant coating [8]. In stomatology, it is used to fill in bone loss after hemisection and radectomy and as a material preventing tooth sensitivity [10]. Limeback et al. reported that HAp restored enamel by creating a new coating on the tooth surface [12], thus providing protection from tooth decay and limiting the adhesion of dental plaque [13]. 

It should be emphasized that the addition of HAp to a composite material affects the mechanical properties of the filling [13,14,15], where the form and amount of the addition are of importance. It was found that an addition of over 30% hydroxyapatite in the form of powder increased the rigidity of the material [15]. It has also been reported that the use of HAp at a nanometric scale improved composites’ mechanical properties [13,14]. In turn, Santos et al. [16] demonstrated that the addition of 3% hydroxyapatite in the form of nanofibers introduced into a TEGDMA/Bis-GMA polymer matrix improved the material strength [16,17]. 

Due to the lack of complex data concerning the effect of hydroxyapatite addition on the key mechanical properties of dental composites, the present study examined the effects of such modification on the hardness and strength of the material, and more precisely, its static properties (compression and bending strength), dynamic properties (impact strength and brittle crack resistance), and utilitarian properties (abrasion resistance). The obtained data can be used to indicate the optimal composition for fillings, offering optimal biological and mechanical advantages. 

## 2. Materials and Methods

The samples were prepared from flow-type commercial composite material (Arkona Flow Art, Niemce, Poland) (*n* = 60) and flow-type material modified with added sintered hydroxyapatite in the form of powder with a grain size of 30 µm at 2% wt. (*n* = 60), 5% wt. (*n* = 60), and 8% wt. (*n* = 60). The commercial composite was composed of bisphenol A diglycidyl ether dimethacrylate, diurethane dimethacrylate, triethylene glycol dimethacrylate, barium–aluminum–silicon glass, titanium dioxide, silica, and camphorquinone. To the weighed amount of the conventional composite, small portions of the filler were added and then manually ground with a pestle in an agate mortar. These samples were kept in polypropylene syringes with a plunger. The HAp used in the work was synthesized using the wet method. The dried HAp grains were fractionated using an LPzE-3e laboratory shaker (MULTISERW-Morek, Brzeźnica, Poland) through a set of three sieves with the following mesh sizes: 0.1 mm, 0.05 mm, and 0.025 mm. The HAp was then introduced into the composite material using a Roti-Speed stirrer (Carl Roth GmbH + Co. KG, Karlsruhe, Germany). This stirrer was used to mix very small samples in micro-tubes. The composites were mixed at 5000 rpm for about 5 min. The mixing was performed in a darkened room under standardized temperature and humidity conditions. All samples were prepared in the shapes of cuboidal beams (*n* = 120) and cylinders (*n* = 120), with sizes according to ISO norms [18,19,20,21,22,23]. The samples were shaped in silicon molds between laboratory glass slides to protect the surface from oxygen inhibition. They were then exposed to a polymerization diode lamp (Elipar S10, 3M ESPE, St. Paul, MS, USA) at 1400 mW/cm^2^ and 450–490 nm for 20 s on each layer, with a thickness of 1 mm. The samples were subjected to various bending, compression, and diametral compression strength tests, in addition to tests of impact strength, hardness, and tribological wear resistance. Before the mechanical tests, the samples were artificially aged at 37 °C in water for 24 h.

In dentistry, the mechanical properties of the materials used for treating teeth and forming prosthetics are important considerations and are often assessed, especially under load conditions. All materials should be biotolerant, durable, aesthetic, and easy to apply.

### 2.1. Bending Strength Test

For the bending strength tests, 10 samples from each group were prepared in the shape of cuboidal beams with the dimensions of 2 mm × 2 mm × 25 mm. The tests were carried out on a multifunctional UMT TriboLab device (Bruker, Karlsruhe, Germany). The traverse shift speed was 0.5 mm/min, with maintained support spacing of 20 mm. The radii of the supports and the mandrel realizing the input function equaled 1 mm.

### 2.2. Compression Strength Test

For the compression strength test, 10 samples from each group were prepared in the shape of cylinders 4 mm in diameter and 6 mm high. The tests were performed on a Walter + Bai testing machine (Walter + Bai AG, Lohningen, Switzerland).

### 2.3. Diametral Compression Strength Test (DTS)

For the diametral compression strength test, 10 samples from each group were used. They were cylindrical in shape, 4 mm in diameter, and 2 mm high. The DTS test was carried out on a universal Zwick/Roell Z020 testing machine (Zwick/Roell, Ulm, Germany), with a traverse shift speed of 1 mm/min.

### 2.4. Impact Strength Test

For the impact strength test, 10 samples from each group were used. They had the shape of cuboidal beams with dimensions of 5 mm × 10 mm × 20 mm. The test was performed on a HIT 5.5p ZwickRoeler hammer drill (Zwick/Roell, Ulm, Germany) with a hammer energy of 5.5 J.

### 2.5. Hardness Measurements

The hardness measurements were conducted on the surfaces of 10 samples from each group. They had the shape of cuboidal beams with dimensions of 5 mm × 10 mm × 20 mm. Five measurements were performed in randomly selected areas. For the measurements, the Shore hardness tester type D (Elcometer Inc, Warren, MI, USA) was used.

### 2.6. Tribological Wear Resistance Test

For the wear resistance test, 10 samples from each group were used. They were cylinders 21 mm in diameter and 2 mm high. The tests were carried out on a Tribometer device (CSM Instruments, Freiburg, Germany) with the installed Tribox program and the following parameters: friction radius of 6.75 mm, speed of 0.05 m/s, load of 1 N, friction path of 100 m, temperature of 25 °C, and in an artificial saliva environment according to Fusayama/Meyer (2 dm^3^ distilled water, 0.8 g NaCl, 0.8 g KCl, 1.59 g CaCl2•2H2O, 1.56 g NaH2PO4•2H2O, 0.01 g Na2S•9H2O, and 2 g urea) [24]. The samples were mounted in a specially designed Teflon holder, into which the artificial saliva was poured. The counter sample in the friction process was a globule made of zirconium oxide 1/8 inch in diameter. The materials’ wear was determined through a measurement of the linear wear in the area of the friction track based on the measurement of the surface roughness with the use of a Hommel Waveline 200 profilometer (ITA, Skórzewo, Poland). The wear was calculated as a volumetric material loss with respect to the path of friction [25,26,27].

All obtained test results were subjected to statistical analysis using Excel 2010 (Microsoft) and Statistica v. 13 software. The distribution of the continuous variables was determined using the Shapiro–Wilk test of normality. The Kruskal–Wallis test was used to compare variables without a normal distribution. Levene’s test was used to determine the variance of normally-distributed variables. For equal variances, ANOVA was used with the Scheffe test post hoc. The assumed significance level was α = 0.05.

## 3. Results

### 3.1. Bending Strength Test

A statistically significant difference was demonstrated in the bending strength (MPa) (*p*-value = 0.0153; Kruskal-Wallis). In addition, significant differences were found between the 0% wt. HAp and 8% wt. HAp samples (*p*-value = 0.0134 post hoc test of multiple comparisons of mean ranks for all trials), with larger values in the 0% wt. HAp samples (Figure 1).

### 3.2. Compression Strength Test

No significant difference was found for compression (MPa) (*p*-value = 0.05; ANOVA); however, significant differences were demonstrated in the HV (*p*-value = 0.0000; ANOVA) between: 0% wt. HAp and 2% wt. HAp (*p*-value = 0.0049) with larger values in the 0% wt. HAp samples, 0% wt. HAp and 5% wt. HAp (*p*-value = 0.0000), with larger values in 0% wt. HAp, 0% wt. HAp and 8% wt. HAp (*p*-value = 0.0000) with larger values in the 0% wt. HAp samples (Figure 2).

### 3.3. Diametral Compression Strength Test (DTS)

No significant difference was found for the DTS (*p*-value = 0.05; Kruskal–Wallis) (Figure 3).

### 3.4. Impact Strength Tests

Significant differences in the impact resistance (J/cm^2^) were noted among groups (*p*-value = 0.0370; Kruskal–Wallis). Significant differences were found between the 2% wt. HAp and 8% wt. HAp samples (*p*-value = 0.0233; post hoc test of multiple comparisons of mean ranks for all trials), with larger values in the 2% wt. Hap sample (Figure 4).

### 3.5. Hardness Measurements

Significant differences in hardness were found among groups (Shore’s) (*p*-value = 0.0044; Kruskal–Wallis). Significant differences were found between the 0% wt. HAp and 8% wt. HAp samples (*p*-value = 0.00218; post hoc test of multiple comparisons of mean ranks for all trials), with larger values in the 0% wt. HAp sample (Figure 5).

### 3.6. Tribological Wear Resistance Test

Significant differences were found in the tribological wear (10^−4^ mm^3^/Nm) (*p*-value = 0.0148; Kruskal–Wallis). Statistically significant differences were found between the 0% wt. HAp and 8% wt. HAp samples (*p*-value = 0.0233; *post hoc* test of multiple comparisons of mean ranks for all trials), with larger values in the 8% wt. HAp sample (Figure 6).

## 4. Discussion

### 4.1. Bending Strength Test

Composite fillings are exposed to bending forces. Therefore, a three-point bending test was carried out. In all cases, the addition of hydroxyapatite caused as much as a twofold drop in the bending strength. The differences are statistically significant. A requirement of the norm ISO 4049, “Dentistry—Polymer-based restorative materials 2009” [18], made for polymer-based fillers restoring hard tooth tissue, is a bending strength of 50 MPa. All the modified composites fulfilled the requirements of the standard, and hence, appear to be suitable for use as fillings in hard tooth tissue defects in this regard. 

### 4.2. Compression Strength

One of the most important properties of filling materials is their compression strength. During chewing, teeth are largely subjected to compression, with mean a pressure equal to about 100–150 N [17,28]. The addition of hydroxyapatite caused a drop in the compression strength proportional to the hydroxyapatite content in the composite. However, the obtained differences are statistically insignificant. Unfortunately, there are no norms determining the minimal value of compression strength for materials of this type. Still, it seems that the obtained compression strengths are acceptable. 

### 4.3. Diametral Compression Strength

Classic tensile tests are not suitable for materials used for the restoration of hard tooth tissue, as these are too expensive and too brittle. Therefore, a diametral compression strength test was used to determine the abilities of a brittle dental filling to counteract the tensile stresses occurring during chewing; this requires much smaller samples and can be used on brittle materials [19]. However, it should be noted that the obtained values are not the same as those obtained in a classic tensile strength test. Still, as some calculations and modeling processes require a tensile strength value, this test is conducted for materials used in tooth fillings. 

It was found that, similar to the other strength properties, the strength decreased as the HAp addition increased. The minimum DTS value accepted by the American Dental Association is 24 MPa, as indicated by norm no. 27, “Resin-based fillings” [18]. Hence, in this regard, all the examined composites appear to be suitable for use in the restoration of hard tooth tissue.

### 4.4. Impact Strength

Significant differences were noted between particular groups for the impact strength tests. The addition of 2% wt. HAp increased the impact strength from about 0.13 J/cm^2^ to about 16 J/cm^2^. However, further increases reduced the impact strength: 0.12 J/cm^2^ for 5% and 0.10 J/cm^2^ for 8% wt. Both values are already below those of the initial material. 

### 4.5. Hardness

An important value for composite materials is their hardness, which determines their ability to counteract deformations. It was found that the addition of hydroxyapatite decreased the hardness of the experimental samples. The commercial samples had a hardness of about 80 ShD; however, this fell to 68 ShD for samples with 8% wt. HAp. The differences between the commercial samples and the modified ones were significant. However, it should be added that hardness is only an auxiliary parameter that provides an indication of the other mechanical and performance values. Although hardness measurements are often conducted in tests due to their easiness and quickness, they cannot be unequivocally translated to the other properties. Despite the noticed fall in hardness, the achieved values are acceptable for polymer-based composite materials used to restore hard tooth tissue. 

### 4.6. Wear Resistance Test

Wear resistance is also an important property of dental fillings. Friction and wear occur during the chewing process, and this is intensified during the breaking up of hard food. The wear resistance largely determines the length of use of the filling. Tests confirm that high values are associated with more rapid wear of the filling [14,15]. The addition of hydroxyapatite to the examined composite lowered the wear resistance: materials with higher HAp addition demonstrated greater wear. While the addition of 2 and 5% wt. HAp demonstrated acceptable wear, 8% wt. HAp had significantly worse wear. Hence, applying amounts over 5% wt. appears to be unjustified. 

Akhtar et al. indicated that HAp may be a promising material as a composite reinforcing filler [29]. The presence of 0.2–1% HAp particles had a noticeable positive effect on the tribological and mechanical properties of the composite. In addition, the morphology and size of the particles were also found to have a large impact on the tribological and mechanical properties of the acrylic resin nanocomposite [29]. Similarly, the morphology of the filler can have a significant effect on the mechanical properties, as nanometer-sized hydroxyapatite is characterized by much lower values than micro-scale hydroxyapatite [30]. Poorzandpoush et. al. indicated that the addition of HAp also improved the mechanical properties of glass ionomers [31]. A content of up to 10% HAp by weight increased the abrasion resistance of Fuji II LC RMGI material.

The deterioration of the mechanical and tribological properties of the composite with HAp could be related to the change in filler to the polymer matrix (matrix) ratio caused by adding extra filler in the form of hydroxyapatite. This increase in the amount of filler resulted in a relative decrease in the silanizing agent, and thus, a deterioration in the bond between the organic polymer matrix and inorganic filler particles, as confirmed previously [32,33,34]. To prevent the deterioration of the mechanical properties when adding hydroxyapatite, its surface should be modified by increasing the addition of pre-adhesive (silanizing) agents. When considering the amount of silanizing additive, one should also consider the fact that the added hydroxyapatite, due to its fragmentation and shape, significantly increased the surface of the dispersion phase. Therefore, when increasing the proportion of the silanizing agent, one should consider not only how much the volume fraction of the dispersion phase increases, but also how much its surface area increases. It was shown that the appropriate selection of silane for the appropriate fraction of the filler improved the mechanical properties of the composite [35]. An alternative may also be the introduction of other additives, which could improve the properties of dental composites by blocking the propagation of composite matrix cracking (e.g., graphene) [36]. However, it should be borne in mind that adding more than 5% of silane to composites causes the surface to become rougher and nonhomogeneous [37,38], which can significantly reduce the wear resistance.

## 5. Conclusions

The addition of hydroxyapatite to a flow-type composite material worsened its mechanical and tribological properties. However, in the case of the 2% wt. and 5% wt. HAp content, the obtained values seem acceptable. Even so, we should not immediately disqualify materials with a higher hydroxyapatite content, as the properties of composites largely depend on various factors, including the degree of refinement of the added filler, the technique of addition, and the activation of the added particle’s surface. Hence, further studies are needed in this area. 

## Figures and Tables

**Figure 1 materials-15-09016-f001:**
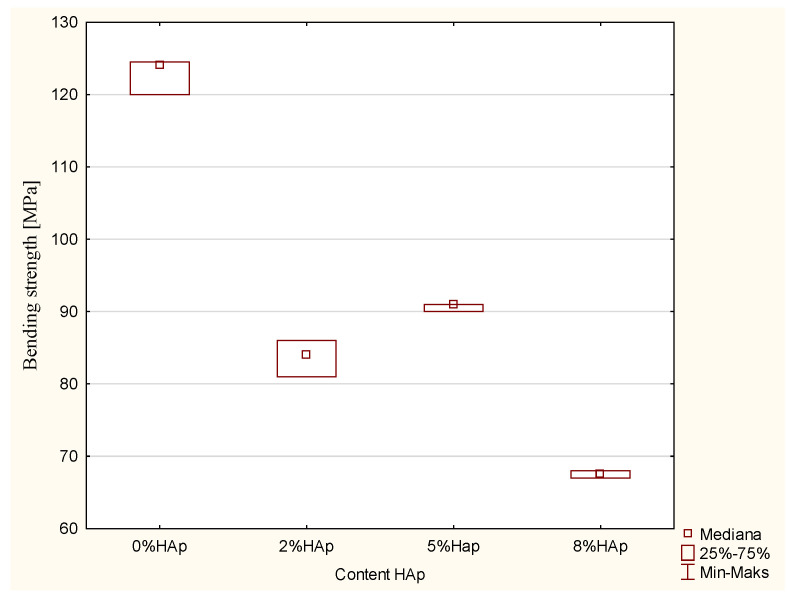
Statistically significant differences in bending strength among examined groups.

**Figure 2 materials-15-09016-f002:**
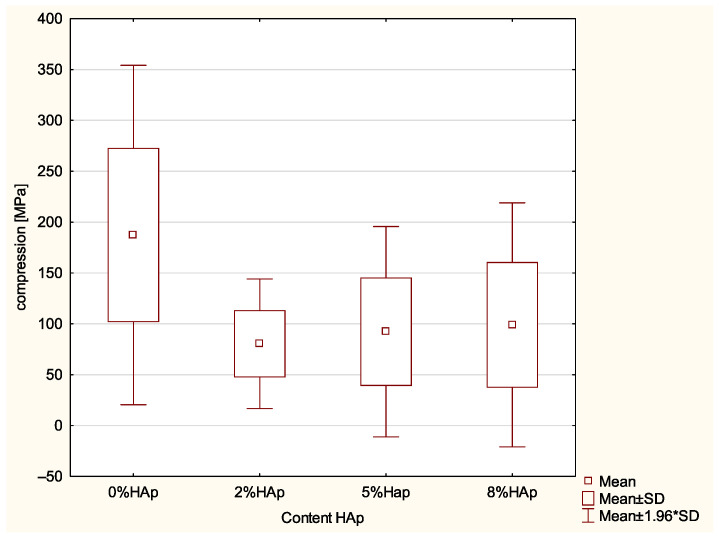
Statistically significant differences in HV among examined groups.

**Figure 3 materials-15-09016-f003:**
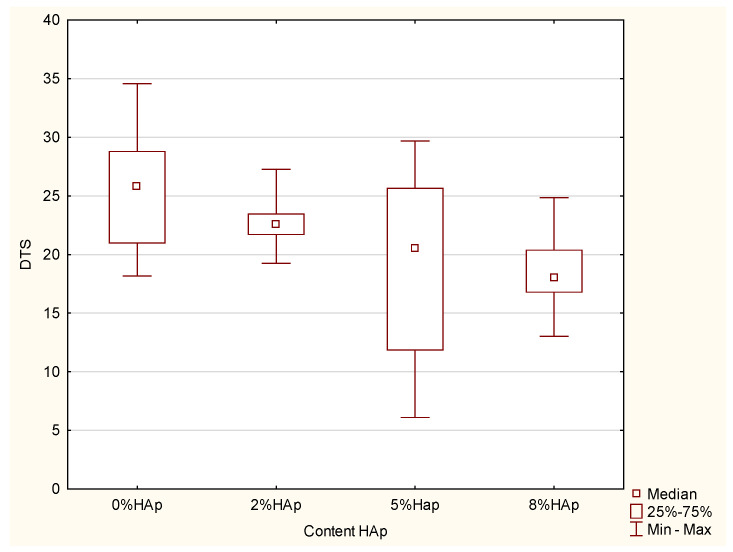
Insignificant differences in DTS among the examined groups.

**Figure 4 materials-15-09016-f004:**
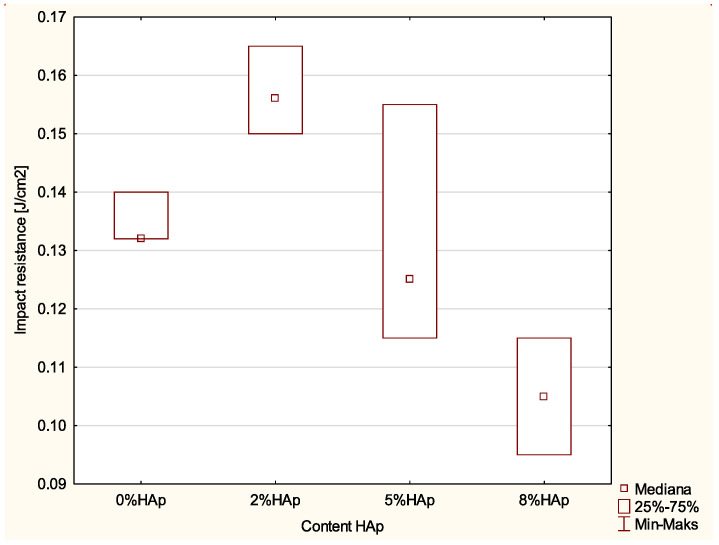
Statistically significant differences in impact resistance among examined groups.

**Figure 5 materials-15-09016-f005:**
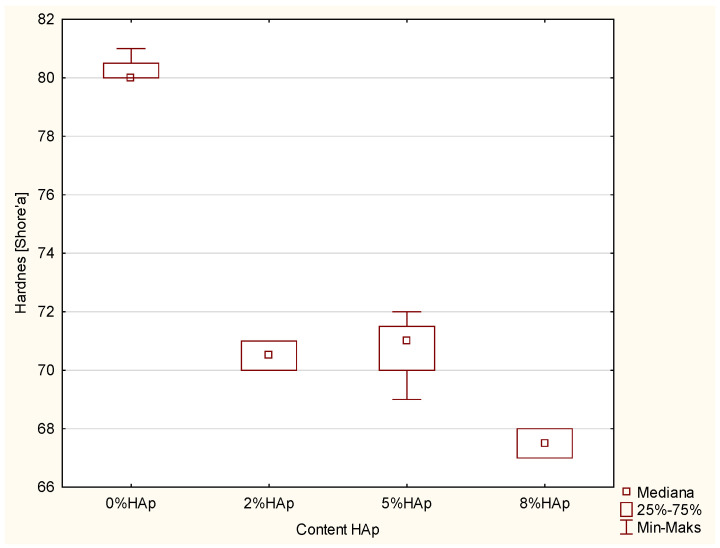
Statistically significant differences in hardness among examined groups.

**Figure 6 materials-15-09016-f006:**
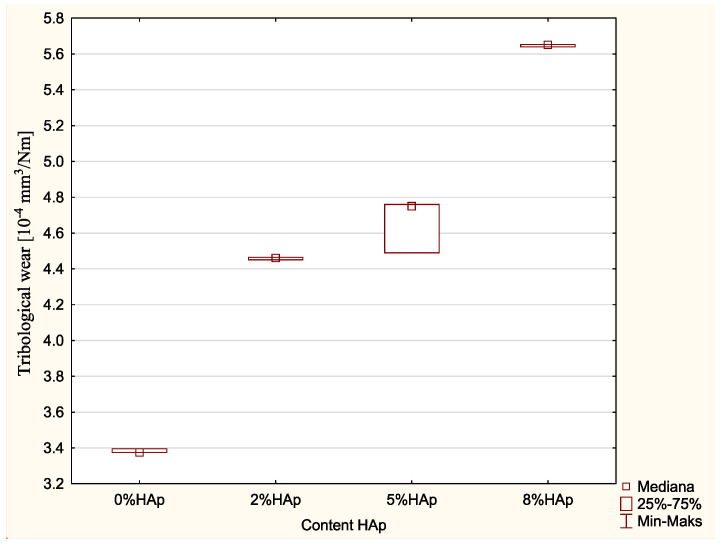
Statistically significant differences in tribological wear among examined groups.

## Data Availability

Not applicable.

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
