# Peer review of "Evaluation of the Effect of the Addition of Hydroxyapatite on Selected Mechanical and Tribological Properties of a Flow-Type Composite"

_materials, 2022, doi:10.3390/ma15249016_

Round 1

Reviewer 1 Report

In this study, static and dynamic mechanical properties of HAp reinforced resin composites were investigated. However, some issues need to be addressed before the manuscript is accepted.

1. Silanization of hydroxyapatite fillers is recommended to improve the bonding between the filler particles and resin matrix.

2. No artificial aging, e.g. 37C for 24 h in water, of the specimens before mechanical testing.

3. The mechanical performance of HAp reinforced resin composites is inferior than the control group except the wear resistance. So, the incorporation of HAp fillers in Bis-GMA/TEGMA resin matrix cannot enhance the mechanical properties of the flow-type resin composites.  

 4. Static and dynamic mechanical properties of HAp reinforced resin composites were investigated. However, the importance of some of these parameters related to the resin composites were not discussed.

Author Response

We would like to submit our article entitled: “Evaluation of the effect of the addition of hydroxyapatite on selected mechanical and tribological properties of a Flow type composite”

The article has been written by Zofia Kula1, Leszek Klimek2, Karolina Kopacz3, Beata Åšmielak4*

  1. Department of Dental Technology, Medical University of Lodz, Pomorska 251, 92-213 Lodz, Poland
  2. Institute of Materials Science and Engineering, Lodz University of Technology, Stefanowskiego 1/15, 90-924 Lodz, Poland
  3. “DynamoLab” Academic Laboratory of Movement and Human Physical Performance, Medical University of Lodz, Pomorska 251, 92-215 Lodz, Poland,
  4. Department of Dental Prosthetics, Medical University of Lodz, Pomorska 251, 92-213 Lodz, Poland

* Corresponding author: [email protected]; Tel.: 0048-603691851

We confirm that neither the manuscript nor any parts of its content are currently under consideration or published in another journal.

All authors have approved the manuscript and agree with its submission to Materials (MPDI)

Comments and Suggestions for Authors

Review 1

Reviewer: In this study, static and dynamic mechanical properties of HAp reinforced resin composites were investigated. However, some issues need to be addressed before the manuscript is accepted.

  1. Silanization of hydroxyapatite fillers is recommended to improve the bonding between the filler particles and resin matrix. 

Answer: Thank you very much for these comments. Due to the fact that these were initial studies, we decided that we would not subject HAp to the silanization process. The next stage of research used fluorapatite, which improves mechanical properties by 0.2%. The composite containing FHA meets the requirements of the standards for dental applications and has the advantage of preventing secondary caries by the release of fluoride, unlike other dental composites. This is also confirmed by the literature:

Arcís, RW.;   López-Macipe, A.; Toledano, M.; Osorio, E.;  Rodríguez-Clemente, R.; Murtra,J.;  Fanovich, MA.;  Pascual, D. Mechanical properties of visible light-cured resins reinforced with hydroxyapatite for dental restoration. Dental Mater. 2002, 1, 49-57.

Reviewer 2 Report

I’ve reviewed the manuscript titled ‘ Evaluation of the effect of the addition of hydroxyapatite on selected mechanical and tribological properties of a Flow type composite’. Following are my comments. 

Authors have made a good effort with reporting the date on the mechanical testing of Flowable resin composite modified by addition of HA particles. 

However, I have some serious reservations about the sample preparation. Authors are encouraged to provide the details on how they incorporated HA in the flow resin. Such manual hand-mixing of HA and dispersion of powder is very likely to deteriorate the mechanical behaviour of resultant sample. Authors are also encourage to explore the literature in relation to the recommended mixing and dispersion of HA in flowable composite to follow a standardize methodology. Additionally, also provided the detailed composition of commercial product Arkona Flow. 

Impact strength measurement are interesting, authors are encouraged to compare the results with similar test methods for resin composite. 

Discussion section should also highlight similar studies which have reported in enhanced mechanical behaviour after nano and micro HA particles. 

More recent and relevant references should be incorporation, author may consider the following paper.

Akhtar, K., Pervez, C., Zubair, N. et al. Calcium hydroxyapatite nanoparticles as a reinforcement filler in dental resin nanocomposite. J Mater Sci: Mater Med 32, 129 (2021). 

Poorzandpoush K, Omrani LR, Jafarnia SH, Golkar P, Atai M. Effect of addition of Nano hydroxyapatite particles on wear of resin modified glass ionomer by tooth brushing simulation. J Clin Exp Dent. 2017

Author Response

We would like to submit our article entitled: “Evaluation of the effect of the addition of hydroxyapatite on selected mechanical and tribological properties of a Flow type composite”

The article has been written by Zofia Kula1, Leszek Klimek2, Karolina Kopacz3, Beata Åšmielak4*

  1. Department of Dental Technology, Medical University of Lodz, Pomorska 251, 92-213 Lodz, Poland
  2. Institute of Materials Science and Engineering, Lodz University of Technology, Stefanowskiego 1/15, 90-924 Lodz, Poland
  3. “DynamoLab” Academic Laboratory of Movement and Human Physical Performance, Medical University of Lodz, Pomorska 251, 92-215 Lodz, Poland,
  4. Department of Dental Prosthetics, Medical University of Lodz, Pomorska 251, 92-213 Lodz, Poland

* Corresponding author: [email protected]; Tel.: 0048-603691851

We confirm that neither the manuscript nor any parts of its content are currently under consideration or published in another journal.

All authors have approved the manuscript and agree with its submission to Materials (MPDI)

Comments and Suggestions for Authors

Review 2

Reviewer: I’ve reviewed the manuscript titled ‘ Evaluation of the effect of the addition of hydroxyapatite on selected mechanical and tribological properties of a Flow type composite’. Following are my comments. Authors have made a good effort with reporting the date on the mechanical testing of Flowable resin composite modified by addition of HA particles. However, I have some serious reservations about the sample preparation. Authors are encouraged to provide the details on how they incorporated HA in the flow resin. Such manual hand-mixing of HA and dispersion of powder is very likely to deteriorate the mechanical behaviour of resultant sample.

Answer: Thank you. The text has been completed.

“The HAp used in the work was synthesized using the wet method. The dried HAp grains were fractionated using a LPzE-3e laboratory shaker (MULTISERW-Morek, Brzeźnica, Poland) through a set of three sieves with the following mesh sizes: 0.1 mm, 0.05mm and 0.025mm. The HAp was then introduced into the composite material using a Roti-Speed stirrer (Carl Roth GmbH + Co. KG, Karlsruhe, Germany). This stirrer is used to mix very small samples in micro-tubes. The composites were mixed at 5,000 rpm for about 5 minutes.”

Reviewer: Authors are also encourage to explore the literature in relation to the recommended mixing and dispersion of HA in flowable composite to follow a standardize methodology. Additionally, also provided the detailed composition of commercial product Arkona Flow.

Answer: Thank you very much for pointing out the literature.

The composition of the composite was added: “The commercial composite was composed of bisphenol A diglycide ether dimethacrylate, diurethane dimethacrylate, triethylene glycol dimethacrylate, barium-aluminum-silicon glass, titanium dioxide, silica and camphorquinone”

Reviewer: Impact strength measurement are interesting, authors are encouraged to compare the results with similar test methods for resin composite.

Answer: Unfortunately, it is not possible to compare the impact strength results, as there are no similar reports in the literature regarding composite materials containing HAp. This is due to the fact that impact tests require quite large samples, and thus high costs. Our research group is the first to undertake such research.

Reviewer: Discussion section should also highlight similar studies which have reported in enhanced mechanical behaviour after nano and micro HA particles. 

Answer: Similar studies have been conducted by authors Akhtar, Pervez and by Zubair et al. These studies indicate that HAp may be a promising material as a composite reinforcing filler. The presence of 0.2-1% HAp particles had a noticeable effect on the tribological and mechanical properties of the dental composite. The research also showed that particle morphology and particle size have a great influence on the tribological and mechanical properties of the acrylic resin nanocomposite.

In order to improve the mechanical properties of glass ionomers HAp was has been added in previous studies (Poorzandpoush K., Omrani LR.; Jafarnia SH.; Golkar P.; Atai M. Effect of addition of Nano hydroxyapatite particles on wear of resin modified glass ionomer by tooth brushing simulation. J Clin Exp Dent. 2017,3, 372-376. doi: 10.4317/jced.53455). It has been found that a content up to 10 wt. NHA increases the abrasion resistance of the Fuji II LC RMGI material.

Zhang and Darvell confirm that the morphology of the filler has a significant impact on the improvement of mechanical properties. Filler with nanometer-size hydroxyapatite has much lower values ​​than those with micro-scale hydroxyapatite.

Reviewer: More recent and relevant references should be incorporation, author may consider the following paper.

Akhtar, K., Pervez, C., Zubair, N. et al. Calcium hydroxyapatite nanoparticles as a reinforcement filler in dental resin nanocomposite. J Mater Sci: Mater Med 32, 129 (2021). 

Poorzandpoush K, Omrani LR, Jafarnia SH, Golkar P, Atai M. Effect of addition of Nano hydroxyapatite particles on wear of resin modified glass ionomer by tooth brushing simulation. J Clin Exp Dent. 2017

Round 2

Reviewer 1 Report

A minor comment:

1. The samples were artificially aged at 37C in water for 24 h or a week? Please provide this information.  

Author Response

We would like to submit our article entitled: “Evaluation of the effect of the addition of hydroxyapatite on selected mechanical and tribological properties of a Flow type composite”

The article has been written by Zofia Kula1, Leszek Klimek2, Karolina Kopacz3, Beata Åšmielak4*

  1. Department of Dental Technology, Medical University of Lodz, Pomorska 251, 92-213 Lodz, Poland
  2. Institute of Materials Science and Engineering, Lodz University of Technology, Stefanowskiego 1/15, 90-924 Lodz, Poland
  3. “DynamoLab” Academic Laboratory of Movement and Human Physical Performance, Medical University of Lodz, Pomorska 251, 92-215 Lodz, Poland,
  4. Department of Dental Prosthetics, Medical University of Lodz, Pomorska 251, 92-213 Lodz, Poland

* Corresponding author: [email protected]; Tel.: 0048-603691851

We confirm that neither the manuscript nor any parts of its content are currently under consideration or published in another journal.

All authors have approved the manuscript and agree with its submission to Materials (MPDI)

Comments and Suggestions for Authors

Review 1

Reviewer: A minor comment: The samples were artificially aged at 37C in water for 24 h or a week? Please provide this information.  

Answer: Thank you very much for these comments. The information has been added: The samples were artificially aged at 37C in water for 24 h.

Reviewer 2 Report

Although authors have some of the previousl suggestion and incorporated chages but the justificatin why the flow type composite material got worsens its mechanical and tribological propertie still requrie further explaination with reference to the available literature. Generally, one of the main vairables is the methods of HA dispersion in the resin composite. Authors should provide recommendation to the reader in relation to the optimise the methodology.

Author Response

We would like to submit our article entitled: “Evaluation of the effect of the addition of hydroxyapatite on selected mechanical and tribological properties of a Flow type composite”

The article has been written by Zofia Kula1, Leszek Klimek2, Karolina Kopacz3, Beata Åšmielak4*

  1. Department of Dental Technology, Medical University of Lodz, Pomorska 251, 92-213 Lodz, Poland
  2. Institute of Materials Science and Engineering, Lodz University of Technology, Stefanowskiego 1/15, 90-924 Lodz, Poland
  3. “DynamoLab” Academic Laboratory of Movement and Human Physical Performance, Medical University of Lodz, Pomorska 251, 92-215 Lodz, Poland,
  4. Department of Dental Prosthetics, Medical University of Lodz, Pomorska 251, 92-213 Lodz, Poland

* Corresponding author: [email protected]; Tel.: 0048-603691851

We confirm that neither the manuscript nor any parts of its content are currently under consideration or published in another journal.

All authors have approved the manuscript and agree with its submission to Materials (MPDI)

Comments and Suggestions for Authors

Review 2

Reviewer: Although authors have some of the previousl suggestion and incorporated chages but the justificatin why the flow type composite material got worsens its mechanical and tribological propertie still requrie further explaination with reference to the available literature. Generally, one of the main vairables is the methods of HA dispersion in the resin composite. Authors should provide recommendation to the reader in relation to the optimise the methodology.

Answer: Thank you very much for these comments. The information has been added:

“The deterioration of the mechanical and tribological properties of the composite with HAp could be related to the change in filler to polymer matrix (matrix) ratio caused by adding extra filler in the form of hydroxyapatite. This increase in the amount of filler resulted in a relative decrease in the silanizing agent, and thus a deterioration in the bond between the organic polymer matrix and inorganic filler particles, as confirmed previously [32-34]. To prevent deterioration of the mechanical properties when adding hydroxyapatite, its surface should be modified by increasing the addition of pre-adhesive (silanizing) agents. When considering the amount of silanizing additive, one should also consider the fact that the added hydroxyapatite, due to its fragmentation and shape, significantly increased the surface of the dispersion phase. Therefore, when increasing the proportion of silanizing agent, one should consider not only how much the volume fraction of the dispersion phase increases, but also how much its surface area increases. It has been shown that appropriate selection of silane for the appropriate fraction of the filler improves the mechanical properties of the composite [35]. An alternative may also be the introduction of other additives, which could improve the properties of dental composites by blocking the propagation of composite matrix cracking (e.g. graphene). However, it should be borne in mind that adding more than 5% of silane to composites causes the surface to become rougher and nonhomogeneous [36-37], which can significantly reduce wear resistance.”

  1. Meena, A.; Mali, H.S; Patnaik, A., Kumar S.R. Comparative investigation of physical, mechanical and thermomechanical characterization of dental composite filled with nanohydroxyapatite and mineral trioxide aggregate. e-Polymers 2017, 4, 311-319, org/10.1515/epoly-2016-031
  2. Lie, N.; Hilton, T.J.; Heintze, S.D.; Hickel, R.; Watts, D.C.; Silikas, N.; Stansbury, J.W.; Cadenaro, M.; Ferracane, J.L. Academy of Dental Materials Guidance-Resin Composites: Part I-Mechanical Properties. Mater. 2017, 33, 880–894
  3. Skapska, A.; Komorek, Z.; Cierech, M.; Mierzwinska-Nastalska, E. Comparison of Mechanical Properties of a Self-Adhesive Composite Cement and a Heated Composite Material, Polymers 2022, 14, 2686, doi.org/10.3390/polym14132686
  4. Orczykowski, W., BieliÅ„ski, D.M., Anyszka, R., Gozdek, T., Klajn, K., Celichowski, G., PÄ™dzich, Z., Wojteczko, A. Fly Ash from Lignite Combustion as a Filler for Rubber Mixes—Part II: Chemical Valorisation of Fly Ash, Materials 2022, 15, 5979, org/10.3390/ma15175979
  5. Vouvoudi, E.C, Sideridou, I.D. Dynamic mechanical properties of dental nanofilled light-cured resin composites: effect of food-simulating liquids. Mech. Behav. Biomed. Mater.2012, 10, 87–96.
  6. Mansour, S.F, El-Dek, S.I, Ahmed, M. Physico-mechanical and morphological features of zirconia substituted hydroxyapatite nanocrystals. Sci. Rep. 2017, 7, 43202, doi.org/10.1038/srep43202
